# Factorial structure of Individual Work Performance Questionnaire (Version 1.0) revisited: Evaluation of acquiescence bias

**Zuleima Santalla-Banderali** [1]* , **Jesús M. Alvarado** [2]

1 School of Psychology, Universidad Espíritu Santo, Samborondón, Ecuador, 2 Psychobiology & Behavioral Sciences Methods Department, Faculty of Psychology, Complutense University of Madrid, Madrid, Spain

☯ These authors contributed equally to this work.
* zuleymasantalla@gmail.com

**Data Availability Statement:** All relevant data are within the manuscript and its Supporting information files.

## Abstract

The objective of this study is to evaluate the factorial structure of the Spanish version of the Individual Work Performance Questionnaire (Version 1.0) and to estimate the impact that acquiescence has on it as response bias. Exactly 500 workers from organizations from different industries, primarily located in Ecuador, participated in this study. The comparison of different models using Confirmatory Factor Analysis proved that when acquiescence is not controlled, evidence leads to the rejection of a one-dimensional—or essentially one-dimensional—structure (bifactor model), thus questioning the initial conceptualization of the construct. But when this response bias is controlled, both the one-dimensional model ($\chi^2$ = 429.608 [223], $p$ < .001; CFI = .974; TLI = .982; RMSEA = .043; SRMR = .063) and the bifactor model ($\chi^2$ = 270.730 [205], $p$ = .001; CFI = .992; TLI = .994; RMSEA = .026; SRMR = .047) show relevant improvement in terms of goodness of fit over the three-correlated-factors model ($\chi^2$ = 537.038 [132], $p$ < .001; CFI = .950; TLI = .942; RMSEA = .079; SRMR = .070). However, the low reliability of the substantive factors of the bifactor model makes the one-dimensional model preferable in applied studies. Finally, the results show how mistakes could be made when concluding on the possible relationships between work performance and other relevant variables, in case acquiescence is not controlled.

## Introduction

The work performance of the members of an organization is one of the key indicators of its performance, success, and sustainability [1–4], and it is as well the basis of multiple organizational processes [5]. Therefore, the explanation of individual work performance is one of the most relevant topics in areas such as industrial and organizational psychology, organizational behavior, management, business, and other related fields [1,2,6–9]. However, in order to attain appropriate explanations of a phenomenon, it is first necessary to validly measure it. Nonetheless, measuring in organizational environments is one of the main shortcomings in organizational behavior research [10].

**Funding:** The author(s) received no specific funding for this work.

**Competing interests:** The authors have declared that no competing interests exist.

Under this context, this research aimed to evaluate the factorial structure of the Spanish version of the Individual Work Performance Questionnaire (IWPQ-Version 1.0) [5] and to estimate the impact that acquiescence response bias has on it. Despite having been developed following a thorough process, IWPQ is a measurement tool that is confronted with empirical evidence, which questions the stability of its factorial structure and even the initial conceptualization of the construct it intends to measure.

Discrepant results regarding the structure of this questionnaire may be associated with acquiescence as response bias. Although this bias frequently occurs when using self-reports [2] that employ Likert-like scales [11–13] with direct and inverse items—as is the case of IWPQ—, its possible incidence has not yet been considered in any of the studies aimed at evaluating the psychometric properties of IWPQ.

## Work performance conceptualization

In organizations, the evaluation of individual work performance is often carried out based on just the level with which people perform the activities that are directly related to the tasks assigned based on the analysis of their job positions [1,7,14,15]. However, work performance goes beyond work itself [16], and it includes a wide range of behaviors that need to be considered to properly measure and understand the construct [3].

Thus, there is some consensus on understanding work performance as the behaviors—rather than the results (i.e., the consequences of behaviors)—of the members of an organization, that are relevant to the achievement of its objectives [5,6,14,17–20], and that are under the control of all individuals [5,20,21]. There is also agreement that the construct is multidimensional [5,19], although there are wide discrepancies in the dimensions proposed by different authors [see 17,22].

Under this context, in order to generate a parsimonious model of job/work performance, Borman & Motowidlo proposed that work performance can be understood as consisting of two large dimensions: (a) task performance -also called "in-role performance"-; and (b) contextual performance -also known as "organizational citizenship behavior", "citizenship performance" or "extra-role performance"- [23]. Apart from those two, other authors have added a third dimension: counterproductive work behaviors [7,8,20,24,25]. Besides, Borman & Motowidlo incorporate adaptive behavior [23], which has been previously pointed out as relevant by different authors [26,27].

Task performance is defined as behaviors that directly or indirectly contribute to the core technical aspects of the organization [1–3,8,23]. That is, the competence level with which individuals perform the core tasks and responsibilities associated with their job [6,8,18–20], which are normally included in job descriptions [14,28,29] or in contracts. Contextual performance is conceived as the individual behaviors that contribute to maintaining and improving the social, organizational, and psychological environment in which task performance takes place [6,19,20], and that go beyond what is contractually prescribed as work objectives [3,4,6,8,14,18,20,23,28,29], but that contribute to achieving the goals of the organization [1,4]. Counterproductive work behaviors are workers' intentional behaviors that are detrimental to the legitimate interests and welfare of the organization or its members [7,8,18–20,25]. Finally, adaptive behavior refers to the extent to which individuals adapt their behavior to changes that occur within their work system or work roles [18,27,28].

These dimensions are conceived as correlated, as proved by some empirical evidence [8,22]. Particularly, a positive and high-magnitude correlation between task and contextual performance has been found [2,14,15,28]. On the other hand, there is a negative and significant correlation between contextual performance and counterproductive work behaviors [7,25]:

employees who frequently perform extra-role behaviors that are beneficial for the organization tend to very rarely perform behaviors that are counterproductive for the organization or its members [18]. A positive relationship between adaptive behavior and task and contextual performance is also expected [18,28], as well as a negative relationship between adaptive behavior and counterproductive work behaviors [18].

Thus, individual work performance is a construct that, in theory, has a hierarchical structure in whose "tip" there is a general factor that is accompanied by several dimensions at the lowest levels, which are correlated, even though they measure different aspects of performance [1,2,8,18].

Following this notion, after controlling different sources of measurement errors, Viswesvaran et al. confirmed the existence of a general factor in the performance evaluations carried out by supervisors and their peers, which explained 60.3% of the total variance [30]. Similarly, Soares de Azevedo Andrade et al. found that the bifactor model, alongside a general factor and two specific ones—task and contextual performance—, was the only one that showed adequate fit indices [2].

The confirmation of the existence of a general factor of work performance implies that it would be justified and significant—psychologically speaking—to combine the scores obtained by the individuals on each dimension in order to procure a general measurement of work performance. However, this does not mean that specific factors lose their importance, since they account for a percentage of the variance that is not explained by the general factor, which can have important theoretical or practical implications [30].

## Measurement of individual work performance: Individual Work Performance Questionnaire

Among the battery of instruments available to measure self-evaluated individual work performance, this study has opted for the Individual Work Performance Questionnaire (IWPQ), version 1.0 [19], adapted to the Spanish language by Ramos-Villagrasa et al. [5].

This instrument has been selected because it was developed following a rigorous process of literature review that supported the selection of the items [29]. Also, because its psychometric behavior has been studied by its original authors [18,19,21,31,32], as well as by other researchers in different countries [4–6,29,33]. Moreover, among its advantages, IWPQ-Version 1.0 includes the three most relevant dimensions of individual work performance: task performance, contextual performance, and counterproductive work behaviors. Additionally, IWPQ is a generic and short instrument that is easy to apply in any organization [4,6,29]. Lastly, the instrument makes it possible to measure self-assessed work performance, which has the advantage of being able to have a performance evaluation in those cases in which it is impossible or extremely difficult to have objective measures of performance [31]. On the other hand, the evaluation that workers make of their own performance could be more accurate because they have more opportunities to observe their own performance than, for example, their supervisors or co-workers, especially considering that the evaluations carried out by supervisors and co-workers can be biased by the halo effect [31]. Despite this, further research is required regarding the psychometric properties of this instrument, since the validation results obtained from different studies cannot be indiscriminately generalized. This is because the relevance of the different performance dimensions and the indicators associated with each of them may depend on the context [29].

With respect to the psychometric properties of IWPQ, all its theoretical dimensions have shown adequate internal consistency indices measured through Cronbach's alpha coefficient (task performance: α between .74 and .89; contextual performance: α between .72 and .91;

counterproductive work behaviors: α between .68 and .89) on studies carried out in different countries, and with versions in different languages [4,6,19,29,31,33].

Regarding the factorial structure of IWPQ, using exploratory structural equation modeling (ESEM), Ramos-Villagrasa et al. found in Spain that the three-correlated-factors model showed a satisfactory fit when correlations were allowed between the errors of two items of the counterproductive work behaviors dimension and between the errors of two items of the contextual performance dimension, whose contents were redundant [5]. Nonetheless, the results from the confirmatory factorial analysis (CFA) have proved that even allowing the correlation between those items, the three-correlated-factors model showed low fit in all the indices used, except for RMSEA [5].

Also, Van der Vaart evaluated in South Africa the psychometric properties of the IWPQ-1.0 version [32] using ESEM and CFA [29]. This author tested three models: (a) a three-correlated-factors model; (b) a model in which the items loaded onto their respective theoretical factors, but those three factors were then allowed to load onto a second-order (performance) factor; and (c) a bifactor model similar to the three-correlated-factors model, except that the items were also allowed to load onto a general (performance) factor [29].

Unlike what has been reported by Ramos-Villagrasa et al. [5], Van der Vaart found that regardless of the analysis performed, all models showed acceptable fit, except for the $\chi^2$ and RMSEA indices [29]. However, as observed by Ramos-Villagrasa et al. [5], when using ESEM, the fit of the three-correlated-factors model considerably improved when the correlation between errors of the same two contextual performance items and between errors of the same two counterproductive work behaviors items was allowed, which Ramos-Villagrasa et al. [5] indeed found to be correlated in their Spanish version of the instrument.

Furthermore, Gabini & Salessi [33] made an Argentine Spanish translation from the Dutch 18-item version of the instrument [31]. After having deleted two items whose factorial load did not reach the established threshold in AFE, CFA showed inadequate fit of the three-correlated-factors model [33]. Adequate fit of this model was only achieved when three other items were deleted, which resulted in a 13-item version [33].

Questions about the three-factor structure of IWPQ also come from the results attained by Daderman et al. using the Swedish translation from the Dutch IWPQ-short version [6]. Using AFE, those authors found four factors: one on which they loaded all counterproductive work behaviors items; another one on which they loaded all task performance items; a third factor on which they loaded five out of the eight contextual performance items; and the fourth one, on which they loaded the three remaining items that in the original longer version (47 items) represented the adaptive work performance dimension [6].

On the other hand, regarding the correlations between the IWPQ factors, there is agreement on the existence of a moderate-to-high magnitude positive correlation between the task and contextual performance dimensions, as well as a smaller-magnitude negative correlation between task performance and counterproductive work behaviors [5,29,33]. However, there are discrepancies in what has been found on different studies regarding the relationship between contextual performance and counterproductive work behaviors: some authors found a low-magnitude negative relationship [29,33]; but others have found a null relationship [5].

Discrepancies regarding the factorial structure of IWPQ call into question its construct equivalence, factorial invariance or metric invariance [13]. On the other hand, the findings of low-magnitude or null correlations between the counterproductive work behaviors dimension and the other two theoretical dimensions could lead to questioning the initial conceptualization of the construct made, which implies that the measurement is conceived either as one-dimensional or as essentially one-dimensional [2,8,18]. This means that its constituent factors are sufficiently related for there to be a general factor that explains the measurement of all

items and/or factors [34]. Additionally, the measurement is conceived as reflective in this conceptualization, so that it is assumed that the items reflect the construct. Therefore, it makes no sense to assume that there may be uncorrelated or theoretically independent factors.

Those IWPQ structural problems may be associated—at least in part—with the measurement method [35], which is a systematic error source that contributes to the variance of the observed scores [11,36], thus making the true scores of those who respond the instrument disagree with the observed score.

The existence of systematic errors due to the measurement method has been presented by Cote and Buckley in the context of organizational psychology [36]. Those authors found in 20 multitrait-multimethod studies, in which either job performance or work satisfaction were measured, that 22.5% of the variance was due to the measurement methods used [36]. Besides, evidence has been found that errors produced by measurement methods can alter the magnitude of the correlation between the scores of different instruments [11,37]. Particularly, Podsakoff et al. found that the magnitudes of the correlations between leader behaviors and variables such as employee performance and leader's perceived effectiveness, as well as the correlations between personality-job performance, job attitudes-organizational citizenship behavior (OCB), participative decision making-work outcomes, organizational commitment-job performance, person and organization fit-job performance, and OCB-performance were significantly greater when the measurements of the constructs came from the same source than when they were obtained from different sources [37].

The effects of the method can be interpreted in terms of biases or response styles (i.e., in terms of the systematic tendency of individuals to respond to the items of an instrument based on aspects other than what each item is supposed to measure) [35]. Response bias can affect the reliability estimates and the factor structure of questionnaires because it distorts the inter-item correlation matrix pattern, which in turn may have a considerable impact on the resulting factor structure [12,38]. A review of the literature indicates that two of the main response biases that occur when measuring non-cognitive characteristics are social desirability (i.e., the tendency for people to present themselves in a generally favorable fashion) and acquiescence [12,13,38,39] or agreement bias [11].

In the present study, we have focused on the acquiescence bias because, on the one hand, the distortion due to this bias seems to be bigger than the distortion due to social desirability and, therefore, its control may be more important when evaluating the psychometric properties of measurement instruments [38]. On the other hand, this bias frequently occurs when used self-reported scales [2] in which Likert-type scales are used [11–13] with direct items (i.e., the items are keyed in such a way that agreement results in a higher scale score) and reverse items (i.e., agreement with an item results in a lower score), as is the case of the IWPQ, and its possible incidence has not yet been considered in any of the studies aimed at evaluating the psychometric properties of IWPQ.

The acquiescence bias is the tendency to systematically respond in the affirmative or to agree with what is stated in the items regardless of their substantive content [11,13,40–43]. It has also been defined as a response style that makes disproportionate use of positive response options [43–46] or higher-score response categories [40]. For example, in the case of the subscale of counterproductive behavior of the IWPQ in which all the items are inverted, acquiescence would be the tendency of people to select the response options identified with high numbers -indicative of a high frequency of occurrence of the behavior and, therefore, worse performance- due to the fact that the person does not pay attention to the substantive content of the item and gets carried away by the fact that in the vast majority of the items of the instrument the response options identified with high numbers are indicative of good performance.

Since acquiescence is not a random error [42], this bias or response style constitutes a source of error variance that distorts self-report measures [40,45,47]. In this regard, evidence has been found that when the measurement instrument contains direct and inverse items, acquiescence causes higher observed scores in cases where most of the items are direct (as is the case of IWPQ). But it results in lower scores if inverse items are predominant within the instrument [11].

Besides, acquiescence can influence the estimates on the reliability [12,47] of the instruments aimed at measuring different constructs under organizational contexts [42]. It can also cause an overestimation of convergent validity [12,42,47], attenuate predictive validity [12], or lead to the underestimation of discriminant validity measurements [42]. Moreover, acquiescence can cause an overestimation of the relationships between constructs whose items are positively worded, and an underestimation of the relationships between constructs whose items are inversely worded [42].

When the contents of the direct and inverse items are not related, acquiescence can generate positive or negative relationships between items where no relationship is expected, causing a distortion of the inter-item correlation matrix and having a deeper impact on the factor structure of the measurement instrument [12,38] of organizational constructs [41,45]. Specifically, Kam & Meyer evaluated the effects of acquiescence on the factorial structure of a scale used to measure job satisfaction, which included direct items (job satisfaction indicators) and inverse items (job dissatisfaction indicators) that had comparable content; therefore, they were supposed to represent a one-dimensional construct in which job satisfaction and job dissatisfaction are the opposite poles of a continuum [41]. Using CFA, the authors found that when acquiescence was statistically controlled, the magnitude of the negative correlation between job satisfaction and job dissatisfaction items was such as to suggest the one-dimensionality of the construct. But when this response bias was not controlled, the magnitude of the negative correlation between the direct and inverse items was attenuated, thus increasing the probability of identifying two dimensions [41].

Similarly, Moors evaluated the impact of acquiescence and extreme response style on the measurement of three leadership styles: transformational, transactional, and *laissez-faire* [45]. In the instrument used, the items that measured transformational and transactional leadership were direct and reflected active behaviors, whereas those that measured *laissez-faire* leadership were inverse and reflected passive behaviors [45]. Each group of items measured different but theoretically related concepts, so negative correlations were expected between the *laissez-faire* items and the transformational and transactional items, as well as positive correlations between the transformational and transactional items [45].

Nevertheless, the author clarifies that when direct and inverse items measure different dimensions (as is the case of IWPQ, in which the items of task and contextual performance are direct/positive, whereas those of counterproductive work behaviors are inverse/negative), it is often observed that negative correlations are of lower magnitude than the positive ones [45]; as previously explained, this has also been found with IWPQ. In such cases, acquiescence could be the reason for observing artificially low negative correlations and artificially high positive correlations [45].

Thus, by using a latent-class confirmatory factor model, Moors evaluated the degree of fit of five models: (a) a three-correlated-factors model; (b) a model that included an acquiescence factor and three substantive factors referring to the three leadership styles; (c) a model in which acquiescence and the extreme response style were controlled; (d) a one-dimensional model without considering any of the two response biases; and (e) a one-dimensional model in which both response biases were controlled [45]. Likewise, this author evaluated whether

the magnitudes of the correlations between the different leadership styles were equaled when acquiescence and the extreme response style were considered [45].

The results proved that, when acquiescence was controlled, the fit of the three-correlated-factors model was greater than that of this same model when this response bias was not controlled; and fit was further improved by also controlling the extreme response style [45]. As for the assumptions about the magnitude of the correlations between factors, they were found to be virtually equal only when both acquiescence and the extreme response style were simultaneously controlled [45]. However, when only one of the two biases was controlled, the negative correlations between *laissez-faire* style and transformational and transactional style continued to be of lower magnitude than the positive correlations between transformational and transactional style [45]. The author explained this result by stating that, since the number of direct items was twice the number of the inverse items, acquiescence partially captured the extreme responses, thus being it confounded with the extreme response bias [45].

Moors also confirmed that response biases can modify the observed relationships between different constructs: when acquiescence was not controlled, a significant relationship between gender and *laissez-faire* leadership was obtained, which ceased to be significant when acquiescence and the extreme response style were simultaneously controlled [45].

However, evidence that controlling acquiescence improves the factorial solution found for an instrument has not been proved in all cases. For instance, Soares de Azevedo Andrade et al. (study 2) used the Self-Assessment Scale of Job Performance (short version) with pairs of positive and negative items to evaluate the influence of acquiescence [2]. For this, they used the Random Intercept Model (RIFA) and incorporated a method factor related to response bias setting out the loads of all items to 1. CFA proved that acquiescence accounted for just 4% of the variance, and controlling it did not improve the fit of the bifactor model (one general factor and two specific factors: task and contextual performance) found when response bias was not controlled [2].

Taking the aforementioned as a starting point, in the particular case of IWPQ, if the scores are contaminated by an "independent" factor of the construct—such as acquiescence—, it is obvious that empirical results that are inconsistent with the theory would be obtained. And this response bias could be obscuring the true structure of the construct, which can certainly be either one-dimensional or essentially one-dimensional (bifactor).

In order to test this assumption, this study has evaluated the factorial structure of the Spanish version of the Individual Work Performance Questionnaire-Version 1.0 [5], estimating the impact that acquiescence has on it. On the one hand, three models in which acquiescence was not considered were tested for this purpose: (a) the classical three-correlated-factors model; (b) a one-dimensional model; and (c) a bifactor model with one general factor and three specific factors. And on the other hand, two other models were tested, in which acquiescence was controlled by isolating, in an independent factor, the variance caused by this response bias, using a confirmatory method based on the Random Intercept Items Factor Analysis (RIFA) [48]: (a) a bifactor model with acquiescence; and (b) a single-factor model with acquiescence.

## Materials and methods

### Sample

This study involved 500 workers (70.6% women) from mostly private organizations (89.2%), primarily located in Ecuador (88.4%), from different sectors (53.3% from the education sector). Participants were between the ages of 17 and 69 ($M = 33.45$; $SD = 10.59$). Most of them had a high educational level (65.1% had a college degree, and 16.2% had completed

**Table 1. Sample characteristics.**

|  |  | Percentage |
|---|---|---|
| Gender | Male | 29.4 |
|  | Female | 70.6 |
| Education level | Elementary and middle school | 0.8 |
|  | Community college degree | 3.4 |
|  | High school | 13.4 |
|  | College degree | 65.1 |
|  | Postgraduate degree | 16.2 |
|  | Other | 1.0 |
| Years of work experience | Less than 1 year | 10.6 |
|  | Between 1 and 5 | 26.3 |
|  | More than 5 years | 63.0 |
| Time in the organization | Less than 1 year | 29.6 |
|  | Between 1 and 5 | 41.3 |
|  | More than 5 years | 29.1 |
| Time in current position | Less than 1 year | 35.4 |
|  | Between 1 and 5 | 42.5 |
|  | More than 5 years | 22.2 |
| Type of contract | Fixed-term or indefinite contract | 71.7 |
|  | Civil servant's contract | 2.9 |
|  | Temporary contract | 10.7 |
|  | Freelance | 8.2 |
|  | Employed without a contract | 5.3 |
|  | Other | 1.2 |
| Type of employee | Part-time | 20.7 |
|  | Full-time | 79.3 |
| Type of organization | Public | 10.8 |
|  | Private | 89.2 |

postgraduate studies). Also, the majority worked full-time (79.3%) under a fixed-term or indefinite contract (71.7%) and had more than five years of work experience (63.0%) (Table 1).

This research was reviewed and approved by the committee designated for it by the Research Center of the Universidad Espíritu Santo in January 2021, which considers in its evaluation the adjustment to the basic ethical norms of research. Following the ethical principles of psychologists and the Code of Conduct of the American Psychological Association [49], participation in this study was voluntary, and subjects were provided a digital informed consent after having received information about: (a) the purpose of the research, expected duration, and procedures; (b) their right to decline to participate and to withdraw from the research once participation has begun without consequences; and (c) whom to contact for questions about the research. Given the characteristics of this research, participating in it did not imply any type of physical or psychological risk for the participants. Nor did it imply subjecting the participants to conditions that could cause them discomfort or that could have adverse effects on them. All data were treated confidentially, and they were not shared with anyone other than the study authors. No specific incentive was used for participation in this study.

## Instrument

The Spanish version of the Individual Work Performance Questionnaire (IWPQ-Version 1.0) [5] was used to measure work performance. Its 18 items evaluate the three dimensions of the

construct: (a) task performance (5 items); (b) contextual performance (8 items); and (c) counterproductive work behaviors (5 items). In this questionnaire, participants were asked to indicate how often they performed each of the behaviors indicated in the items in the last three months, following a Likert scale.

The items of the subscales of task and contextual performance are direct/positive. This means that the more frequently the indicated behaviors are performed, the better the work performance. On the contrary, the items of the counterproductive work behaviors subscale are inverse/negative. This means that the more frequently those behaviors are performed, the worse the work performance. Thus, as it is common in applied research, the scale has been just partially balanced, since few items are worded opposite to the others [44,47,50], whereas the inverse items have not been intended to control acquiescence, therefore, their meaning is not the opposite to that of the directly worded items.

In the IWPQ version used in this research, the original five-interval scales (seldom, sometimes, frequently, often, and always, for the task and contextual performance items; and never, seldom, sometimes, frequently, and often, for the counterproductive work behaviors items) were modified by using six-interval scales with the same response options for all items (1 = never/almost never– 6 = almost always/always), and by labeling just the extremes of the categories. This was done because people often find it very difficult to determine the difference between "frequently" and "often". In fact, 25% of the people who participated in the pilot study made by Koopmans et al. pointed out that the distinctions between the response categories were unclear, especially the distinction between "frequently" and "often" [19].

On the other hand, only the extremes of the scales have been tagged, since it has been observed that tagging all response options can lead to greater acquiescence bias [51]. Finally, six-interval scales were chosen, since increasing the number of intervals of the scales increases the probability that data distribution fits to normal distribution [52]. This may also contribute to improve the reliability of the instrument [52,53] and its convergent validity [53], and it also increases information retrieval [51,53].

## Data analysis

A Confirmatory Factor Analysis (CFA) was conducted to fit different measurement models: (a) a three-correlated-factors model (3F); (b) a single-factor model (1F); (c) a bifactor model with one general factor and three specific factors (BiF); (d) a bifactor model with three substantive factors, one general factor, and one acquiescence factor (BRIFA); (e) a single-factor model with one acquiescence factor (1F+AQ).

The Lavaan package [54] in R Project for Statistical Computing (version 4.1.0) and the robust diagonally weighted least squares estimator (RDWLS)—using a polychoric correlation matrix—were used for CFA. This estimator is recommended for analyzing samples with a small-to-moderate number of observations with ordinal data, such as that yielded by Likert-type items [55,56]. The $\chi^2$ tests, the Root Mean Square Error of Approximation (RMSEA), the Standardized Root Mean Square Residual (SRMR), the Comparative Fit Index (CFI), and the Tucker-Lewis Index (TLI) were used for fit evaluation of the CFA model. The following criteria were used to evaluate model fit: RMSEA < 0.08, SRMR $\leq$ 0.08, CFI and TLI $\geq$ 0.95 [57].

A RIFA method was used in the two models (BRIFA and 1F+AQ) in which acquiescence was controlled. In order to be estimated, the variance of the random intercept was set free in RIFA, with non-null values reflecting the presence of systematic variance associated with acquiescence. Including a random intercept is mathematically equivalent to adding an additional orthogonal acquiescence factor, in which all loadings on the factor are constrained to be

equal and positive [40]. Specifically, in this case the random intercept has been estimated by fixing a load of 1 in all items before reversing the negative items [47].

The usefulness of RIFA has been confirmed with partially-balanced multidimensional scales [40]. One of its advantages is that it is easy to implement because it only requires adding an additional orthogonal factor, in which all items have the same weight. And it is also robust to the violation of the assumption of tau-equivalence in the acquiescence factor loadings [12,40]. Finally, it has been found that the application of RIFA to investigate the factorial structure leads to clearer factor structures and improves model-to-data-fit [12]. These characteristics make this method a precise tool to control acquiescence in confirmatory analyzes of the latent structure of questionnaires and scales [47].

SPSS 25.0 was used for the descriptive analysis of the IWPQ-Version 1.0 items.

## Results

### Descriptive analysis

The analysis of the frequency distribution of each item showed that there was a ceiling effect in all items of the task and contextual performance subscales since more than 15% of the participants indicated that, in the last three months, they had almost always/always performed the behaviors indicated in the items of those two subscales. On the other hand, there was a floor effect in all items of the counterproductive work behaviors subscale, since more than 15% of the participants indicated that, in the last three months, they had never/almost never performed the behaviors indicated in the items of that subscale (Table 2).

In fact, all items presented a distinctive negative skewness, so that the scores tended to cluster towards the higher values—indicative of high work performance—, with medians between 5 and 6 for the task and contextual performance items, and medians between 4 and 5 for the counterproductive work behaviors items (Table 2). Nonetheless, skewness was considerably higher for the task and contextual performance items (between -0.842 and -1.641), compared to that of the counterproductive work behaviors items (between -0.051 and -0.629).

This less asymmetric distribution of the counterproductive work behaviors items is due to the fact that a greater percentage of people selected options 4, 5, and 6 from the scales, which were indicative of a greater frequency of occurrence of those behaviors, and which were assigned 3, 2, and 1 points, respectively, as seen in Table 2. Finally, the distributions of the task and contextual performance items were leptokurtic, whereas those of the counterproductive work behaviors items were platykurtic.

These differences between the distributions of the direct items (task and contextual performance) and the inverse items (counterproductive work behaviors) suggest that acquiescence response bias may be occurring.

### CFA model fit

As it can be seen in Table 3, the classical three-correlated-factors model (3F) presents adequate RMSEA, SRMR, and CFI fit indices. However, it shows inadequate goodness of fit in TLI and in $\chi^2$. The inadequate goodness of fit, according to $\chi^2$, could be due to the fact that this test is very sensitive to detect model misfit when working with large samples [58]. As it can be seen, there is a strong correlation between task and contextual performance ($r = .859$, $p < .0001$). However, the correlations between those factors and counterproductive work behaviors are either low or null: counterproductive work behaviors-task performance: $r = -.189$, $p < .001$; counterproductive work behaviors-contextual performance: $r = -.052$, $p = .259$ (Fig 1).

The low correlation observed between counterproductive work behaviors and the other two factors leads to the rejection of a possible one-dimensional or essentially one-dimensional

**Table 2. Frequency distribution and descriptive statistics of each item of IWPQ-Version 1.0.**

| Item[a] | Frequency distribution | | | | | | M | Md | SD | As | K |
|---|---|---|---|---|---|---|---|---|---|---|---|
| | 1 | 2 | 3 | 4 | 5 | 6 | | | | | |
| **Task performance (TP)[b]** | | | | | | | | | | | |
| TP1 | 0 | 2.6 | 7.2 | 13.8 | 35.8 | 40.6 | 5.05 | 5.00 | 1.025 | -1.046 | 0.538 |
| TP2 | 0.2 | 0.4 | 1.6 | 8.4 | 31.2 | 58.2 | 5.45 | 6.00 | 0.779 | -1.641 | 3.579 |
| TP3 | 0.6 | 1.2 | 2.4 | 12.4 | 33.6 | 49.8 | 5.27 | 6.00 | 0.910 | -1.481 | 2.739 |
| TP4 | 0.2 | 0.2 | 2.6 | 8.0 | 36.1 | 52.9 | 5.38 | 6.00. | 0.790 | -1.466 | 2.845 |
| TP5 | 0.6 | 2.4 | 3.2 | 16.2 | 40.5 | 37.1 | 5.04 | 5.00 | 0.980 | -1.264 | 2.067 |
| Total | | | | | | | 5.24 | 5.4 | 0.645 | -0.957 | 0.641 |
| **Contextual performance (CP)[b]** | | | | | | | | | | | |
| CP6 | 1.6 | 1.2 | 6.6 | 13.8 | 31.6 | 45.2 | 50.9 | 5.00 | 1.091 | -1.415 | 2.124 |
| CP7 | 1.0 | 2.2 | 3.8 | 15.0 | 31.4 | 46.6 | 5.13 | 5.00 | 1.060 | -1.422 | 2.131 |
| CP8 | 0.2 | 0.4 | 2.4 | 12.6 | 32.1 | 52.3 | 5.32 | 6.00 | 0.844 | -1.279 | 1.820 |
| CP9 | 0.6 | 1.2 | 5.0 | 13.4 | 33.8 | 46.0 | 5.17 | 5.00 | 0.979 | -1.331 | 1.925 |
| CP10 | 0.6 | 1.0 | 4.8 | 18.4 | 33.7 | 41.5 | 5.08 | 5.00 | 0.989 | -1.108 | 1.307 |
| CP11 | 3.6 | 3.8 | 5.2 | 15.2 | 29.4 | 42.8 | 4.91 | 5.00 | 1.303 | -1.378 | 1.417 |
| CP12 | 2.0 | 2.4 | 9.2 | 16.8 | 32.4 | 37.2 | 4.88 | 5.00 | 1.191 | -1.103 | 0.858 |
| CP13 | 1.4 | 2.2 | 5.6 | 12.0 | 29.3 | 49.5 | 5.15 | 5.00 | 1.096 | -1.494 | 2.125 |
| Total | | | | | | | 5.09 | 2.25 | 0.715 | -0.842 | 0.225 |
| **Counterproductive work behaviors (CB)[b]** | | | | | | | | | | | |
| CB14 | 8.0 | 9.6 | 9.8 | 13.9 | 22.9 | 35.5 | 4.44 | 5.00 | 1.654 | -0.627 | -0.263 |
| CB15 | 14.4 | 10.0 | 10.6 | 11.2 | 20.0 | 33.8 | 4.13 | 5.00 | 1.824 | -0.537 | -1.159 |
| CB16 | 7.8 | 10.4 | 11.8 | 16.0 | 20.6 | 33.3 | 4.32 | 5.00 | 1.637 | -0.629 | -0.834 |
| CB17 | 16.4 | 16.0 | 16.8 | 13.4 | 17.2 | 20.2 | 3.60 | 4.00 | 1.759 | -0.051 | -1.338 |
| CB18 | 15.4 | 12.4 | 14.2 | 14.0 | 14.0 | 29.9 | 3.87 | 4.00 | 1.827 | -0.252 | -1.346 |
| Total | | | | | | | 4.07 | 4.20 | 1.289 | -0.358 | -0.581 |

[a]In order to facilitate the comparison between different translations and adaptations, the items have been grouped in scales based on the current 18-item English version of IWPQ [19].

[b]The items of the task and contextual performance dimensions were scored directly (never/almost never = 1 –almost always/always = 6), whereas those of counterproductive work behaviors were scored inversely (never/almost never = 6 –almost always/always = 1). This means that in all subscales a higher score indicates better job performance.

structure. This is verified by the unacceptable goodness of fit of the one-dimensional model (1F, Table 3) and the non-convergence of the bifactor model.

When an acquiescence factor is included in both the one-dimensional model (1F+AQ) and the bifactor model (BRIFA), a substantial improvement in goodness of fit with respect to the three-correlated-factors model (3F) is observed in all considered indices (Table 3). However, the BRIFA model shows a greater fit than the single-factor model (1F+ AQ), since the value of

**Table 3. Goodness of fit indices for the competing models.**

| Model | $\chi^2$ (df) | p | CFI | TLI | RMSEA (90% CI) | SRMR |
|---|---|---|---|---|---|---|
| 1F | 2,761.999 (135) | < .001 | .674 | .630 | .199 (0.193–0.206) | .143 |
| 3F | 537.038 (132) | < .001 | .950 | .942 | .079 (0.072–0.086) | .070 |
| 1F+AQ | 429.608 (223) | < .001 | .974 | .982 | .043 (0.037–0.050) | .063 |
| BRIFA | 270.730 (205) | .001 | .992 | .994 | .026 (0.016–0.033) | .047 |

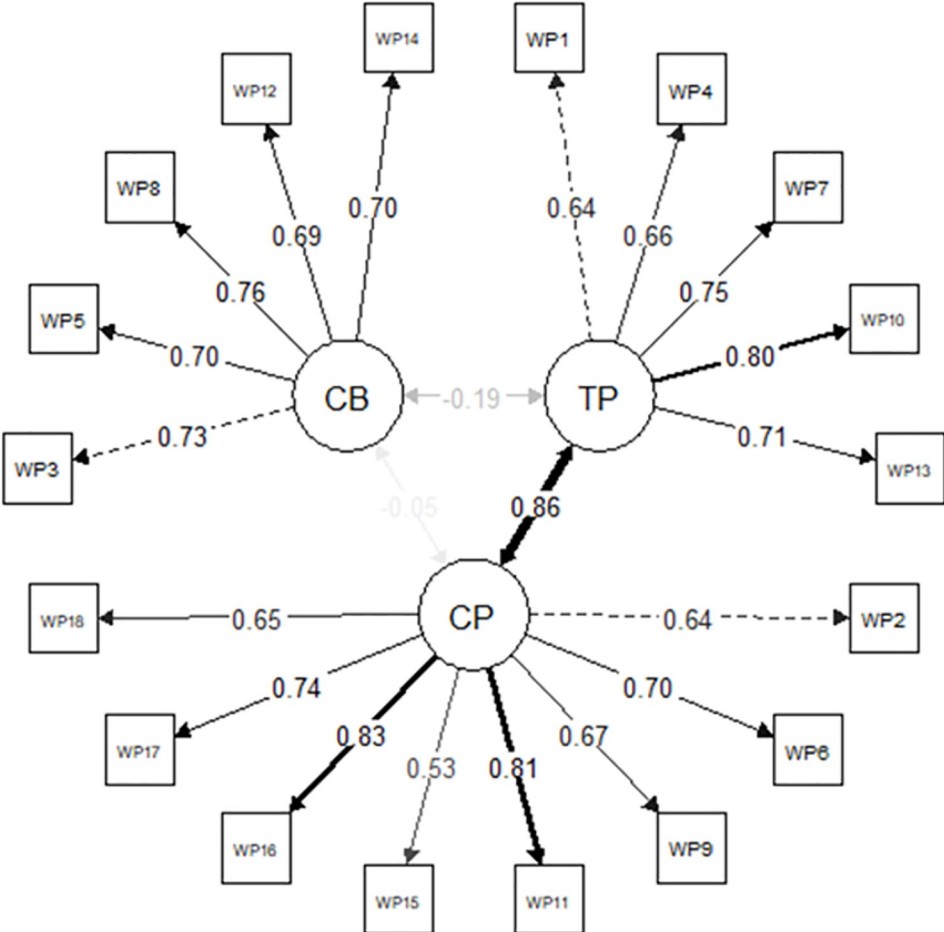

**Fig 1. Path Diagram of 3F model.** CB = counterproductive work behaviors, TP = task performance, CP = contextual performance.

$\chi^2$ was significantly lower in the BRIFA model than in the 1F+ AQ model (Difference of $\chi^2$ = 189.53; $df$ = 18; $p < .001$).

Moreover, in the BRIFA model, the load of the counterproductive work behaviors items on the general factor is acceptable and in the expected direction. Also, the weight of the acquiescence factor is .394, and the three substantive factors are justified, although the reliability of those factors is low (Table 4). On the other hand, in the 1F+AQ model, the load of the counterproductive work behaviors items is adequate, and the weight of the acquiescence factor increases to .470 with respect to that obtained on BRIFA (Table 4).

In order to illustrate the implications that considering or not acquiescence bias may have on the conclusions reached, this study has evaluated the extent to which variables such as sex (SEX), age (EDA), education level (NIV), time in the organization (TIMEINO), time in current position (TIMEINA), years of work experience (EXP), type of organization in which respondents work (TIP: public or private), and type of employee (JOR: part-time or full-time) affect the scores of the substantive domain (work performance [WP]) and the scores of acquiescence (AQ) (Fig 2). A multiple-indicator multiple-cause (MIMIC) structural equation model (SEM) was used for this purpose.

**Table 4. Estimated factor loading for competing models.**

| Items | 1F | 3F | | | 1F+AQ | | BRIFA | | | | |
|---|---|---|---|---|---|---|---|---|---|---|---|
| | WP | TP | CP | CB | AQ | WP | AQ | WP | TP | CP | CB |
| WP1 | .614 | .644 | | | .470 | .456 | .394 | .513 | .332 | | |
| WP4 | .618 | .657 | | | .470 | .441 | .394 | .552 | -.281 | | |
| WP7 | .705 | .752 | | | .470 | .533 | .394 | .641 | -.031 | | |
| WP10 | .753 | .799 | | | .470 | .585 | .394 | .676 | .163 | | |
| WP13 | .673 | .713 | | | .470 | .516 | .394 | .595 | .182 | | |
| WP2 | .621 | | .641 | | .470 | .440 | .394 | .462 | | .184 | |
| WP6 | .677 | | .703 | | .470 | .480 | .394 | .550 | | .108 | |
| WP9 | .650 | | .675 | | .470 | .450 | .394 | .410 | | .379 | |
| WP11 | .791 | | .805 | | .470 | .652 | .394 | .619 | | .337 | |
| WP15 | .503 | | .534 | | .470 | .275 | .394 | .181 | | .488 | |
| WP16 | .813 | | .828 | | .470 | .661 | .394 | .581 | | .466 | |
| WP17 | .715 | | .741 | | .470 | .515 | .394 | .412 | | .552 | |
| WP18 | .626 | | .648 | | .470 | .429 | .394 | .401 | | .336 | |
| WP3 | -.308 | | | .735 | .470 | -.622 | .394 | -.495 | | | .404 |
| WP5 | -.158 | | | .701 | .470 | -.459 | .394 | -.298 | | | .514 |
| WP8 | -.316 | | | .759 | .470 | -.643 | .394 | -.539 | | | .366 |
| WP12 | -.136 | | | .691 | .470 | -.437 | .394 | -.248 | | | .587 |
| WP14 | -.223 | | | .696 | .470 | -.528 | .394 | -.403 | | | .407 |
| Compose Reability | .843 | .839 | .885 | .841 | .836 | .863 | .768 | .844 | .171 | .543 | .569 |

Adequate goodness of fit was observed in all indices, except for $\chi^2$, which is sensitive to sample size ($\chi^2$ [351] = 507,167, $p < .001$; CFI = .977; TLI = .990; RMSEA = .032 [0.025–0.038]; SRMR = .063).

Regarding the prediction of work performance scores, it was found that the model explained 12.1% of the variance ($R^2$ = .121), obtaining a mean effect size ($r$ = .35), according to the criteria of Cohen [59] and Schäfer & Schwarz [60]. Specifically, it was found that work performance scores significantly increased as respondents' age and years of work experience increased.

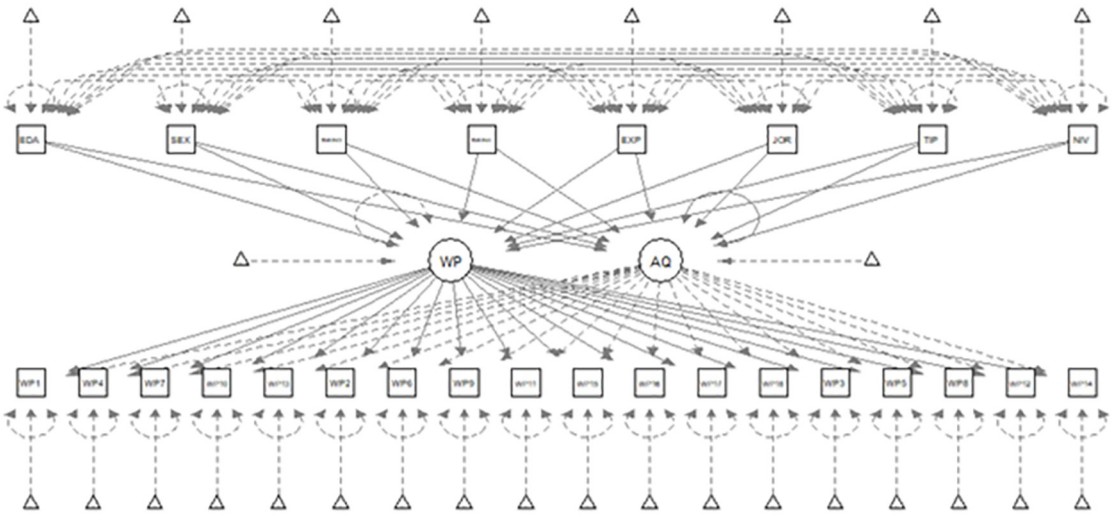

**Fig 2. Predictive model tested.**

**Table 5. Results of the predictive model tested.**

| Predictor | Work performance | | Acquiescence | |
|---|---|---|---|---|
| | β | p | β | p |
| Age | .188 | .001 | -.053 | .411 |
| Years of work experience | .180 | .005 | -.120 | .087 |
| Time in the organization | -.201 | .038 | .073 | .443 |
| Type of organization | .122 | .017 | .033 | .531 |
| Gender | -.008 | .867 | -.104 | .038 |
| Time in current position | .025 | .798 | .200 | .031 |
| Type of employee | .088 | .084 | .067 | .224 |
| Education level | -.010 | .835 | -.007 | .881 |

However, work performance scores were lower in respondents who had more time working in the organization. Moreover, work performance of workers from private organizations was higher than that of those from public organizations. The rest of the variables did not significantly predict work performance (Table 5).

With respect to acquiescence, the model explained 6.7% of the variance ($R^2$ = .067), obtaining a small effect size ($r$ = .26), according to Schäfer & Schwarz [60]. It was found that acquiescence scores were significantly predicted by (a) sex: the tendency to acquiescently respond was significantly higher in men than in women; and (b) time in current position: acquiescence scores increased as respondents' time in current position was longer. The rest of the variables did not significantly predict acquiescence (Table 5).

These results show that if acquiescence is not controlled, errors could be made when interpreting the results obtained regarding the relationships between the work performance construct and other potentially relevant variables. In this study, work performance did not significantly differ depending on sex, although men showed greater acquiescence. In a hypothetical situation, in which the total score in IWPQ was used without controlling this response bias, men could obtain lower total scores if they had lower scores than women on the counterproductive work behaviors items.

This could erroneously lead to the conclusion that men's performance is lower than women's, when in fact, the reason to this is that men tend to select higher response options—which indicate worse performance—more often than women throughout the inverse items of the instrument. The same applies to time in current position. Work performance does not significantly vary depending on this variable. Nonetheless, respondents who have been working for longer in the same position show greater acquiescence bias. Therefore, they could obtain lower total scores than those who have been working for less time in their current positions.

Indeed, this could lead to the conclusion that those who have been working for longer in their current positions have worse performance, as it has been shown when working with the total score on the instrument without controlling acquiescence. However, what actually happens is that those people are less careful when responding and tend to select higher response options—which indicate worse performance—more often than those who have been working for less time their current positions.

## Discussion

The purpose of this study was to evaluate the factorial structure of the Spanish version of the Individual Work Performance Questionnaire-Version 1.0 [5], and to estimate the impact that

acquiescence as response bias has on it, considering a sample of workers from mainly Ecuadorian organizations.

Firstly, the results obtained from this study have shown that, in general terms, participants responded that they frequently performed behaviors that are indicative of high work performance. This was indeed expected, since people tend to evaluate themselves favorably and overestimate their own performance when responding self-report measures [10,31,61]. However, considering the data distribution for each IWPQ item, a negative skewness of smaller magnitude was found in the distributions of the counterproductive work behaviors items, compared to that of the task and contextual performance items. Thus, a higher percentage of people who selected options 4, 5, and 6 from the scales of the counterproductive work behaviors items— indicative of greater frequency of occurrence of such undesirable behaviors—was observed. What may at first seem contradictory is a first indicator that people may have responded acquiescently (i.e., tending to make disproportionate use of higher response categories). This response bias frequently occurs when using self-report measures [2] made up of Likert scales [11–13], which contain direct and inverse items, as is the case of IWPQ.

Regarding the factorial structure of IWPQ, the results herein have shown that the three-correlated-factors model presented acceptable fit. This agrees with what has been previously reported in South Africa [29], where it was found a good fit for this model, especially when correlation between the errors of two of the contextual performance items and between two of the counterproductive work behaviors items was allowed. However, these results differ from those obtained in Spain [5], where it was found that this model had inadequate fit, even when correlation between the items whose errors were also correlated in the study from South Africa [29]. They differ as well from those found in Argentina [33], where was achieved adequate fit on the three-correlated-factors model only when five out of the 18 items of the instrument were deleted.

Despite the acceptable fit of the three-correlated-factors model found in this study, the model turns out to be conceptually conflicting when analyzing what has been observed regarding the correlations between the factors. In this regard, a positive high-magnitude correlation was found between the task and contextual performance factors; just as expected. This fully agrees with what has been reported about IWPQ in countries such as Spain [5], South Africa [29] and Argentina [33], as well as with what has been found with other measurement instruments in Brazil [2], Australia [26], and India [28]. In fact, the magnitude of the correlation found on this study was so high that it may be difficult to discern between those two factors [1,18].

This marked overlap between those two dimensions could in part be due to the criteria that are commonly used to differentiate both dimensions (i.e., whether the behavior is a formal part of the respondent's role, appears in the job description, or is recognized as part of the formal reward system of the organization) [20]. Those criteria can be problematic because the same behavior can be task/in-role or rewarded in one organization or in some jobs, but they can also be contextual/extra-role in another organization or in other jobs [18,20].

On the other hand, task and contextual performance perhaps contribute to the single variance of work performance because supervisors combine aspects associated with contextual performance with aspects associated with task performance when evaluating the global performance of their subordinates. This way, either both supervisors and subordinates end up considering that the behaviors associated to both dimensions are part of the same dimension, or the organization includes—either explicitly or implicitly—contextual behaviors as part of the task behaviors [1,14,23,62].

Nevertheless, the correlation between counterproductive work behaviors and task performance was negative, although low-magnitude. This agrees with what has been reported by

different researchers [25,33], but disagrees with what was expected based on what has been found by others authors who obtained higher-magnitude correlations [5,29]. Furthermore, the correlation between counterproductive work behaviors and contextual performance was null. This agrees with what has been reported in Spain [5], but disagrees with what has been observed in other research [7,29,33].

The low magnitude of the relationship between counterproductive work behaviors and task performance, and the null relationship between counterproductive work behaviors and contextual performance calls into question the initial conceptualization of the construct as consisting of a general factor, accompanied by several correlated dimensions at the lowest levels of the hierarchy [2,8,18]. Such conceptualization implies that it is assumed that although each of the dimensions measure different aspects of work performance, they can also be integrated into a global measurement of the construct [1]. Thus, the measurement is supposed to be reflective. Therefore, it is theoretically inconsistent to assume that factors independent from each other can come up from the construct.

Empirical support for the initial conceptualization of the work performance construct required confirmation that the measurement was one-dimensional; or at least an essentially one-dimensional structure had to be obtained in case different content domains were differentiated, thus obtaining adequate fit indices for the bifactor model. However, the results obtained from this study have shown that, when acquiescence was not considered, the one-dimensional model clearly did not fit, whereas the bifactor model did not converge. These results differ from what has been reported by authors who found that the bifactor model adequately fit to the data [2,29].

Nevertheless, when controlling acquiescence, both the one-dimensional model and the essentially one-dimensional model (BRIFA) showed goodness of fit, although that of BRIFA was slightly higher than that of the one-dimensional model. This is consistent with what has been reported by Viswesvaran et al. who, after controlling different response biases, confirmed the existence of a general factor when performance was evaluated by supervisors and peers [30].

In summary, in our study, we found that when acquiescence is not controlled, the three-correlated-factors model presented an acceptable fit; but, the model turns out to be conceptually conflicting when analyzing what has been observed regarding the correlations between the factors, and when considering the low reliability found on the specific factors.

At first, these results would validate the relevance of the three content domains of the construct, which leads to the appropriateness of using the scores of the factors. If the instrument is conceived as multidimensional, the use of the total score would not be admissible, unless essential unidimensionality is demonstrated; as occurred in the present study when controlling acquiescence. In this case, we found that the one-dimensional model and the essentially one-dimensional model showed goodness of fit. This makes it legitimate and advisable to use the total score obtained on the measurement [34], which is more useful and parsimonious in organizational contexts and applied studies.

These analyses show that acquiescence can alter the dimensionality and the factorial structure [12,38], which has already been verified in organizational contexts for constructs such as job satisfaction [41] and leadership styles [45]. Thus, the analyses herein solve a theoretical debate, which in its three-factor solution would erroneously lead applied researchers to use the instrument as a multidimensional one, instead of considering it as it is: a one-dimensional measurement tool from which extracting the acquiescence factor is necessary for adequate evaluation.

Besides, this study illustrates how, if acquiescence is not controlled, errors could be made when concluding on the possible relationships between work performance and other relevant

variables. Unquestionably, those errors would have relevant and practical implications, not only in terms of the conclusions that can be made on the invariance of the measurement tool, but also in the sense that such errors can lead to incorrect decision-making, thus harming or unfairly benefiting one over the other, and consequently affecting multiple organizational processes.

Thus, our results allow us to conclude that when the dimensionality of a job performance measure is evaluated, it is essential to use some correction method that allows response biases to be controlled, especially acquiescence. Otherwise, distorted factorial structures could be obtained that would lead to errors in the estimations of the scores, and the external validity of the measure [38,63].

Regarding how to control the acquiescence bias, a procedure that can be used when the scale does not contain reverse items involves using the instrument to be evaluated and an additional questionnaire that contained direct and reverse worded items. An index is calculated that evaluates the degree to which people tend to respond acquiescently, and this index allows the variance due to acquiescence on an independent dimension to be isolated. The overall model should be tested using confirmatory factor analysis [64].

Now, the traditional way of trying to control acquiescence bias consists in developing the measuring instrument in such a way that it includes both direct and reversed items that have semantically opposite meanings so that they measure opposite poles of the same theoretical construct. Unfortunately, this strategy does not ensure that the means and covariance structure are free from acquiescence bias [47].

Due to the above, several "a posteriori" methods have been developed that allow for eliminating the impact of acquiescence using statistical procedures [63]. There are procedures based on an exploratory factor analytic approach [65] that can be used in cases of fully balanced scales -half of the scale items are worded in the opposite direction to the other half of the items-.

There are also procedures for quasi-balanced scales -the scale has the same number of positive and negative items but with unequal saturations- or unbalanced scales -only a few items in the scale are worded in the opposite direction to the others, or the scale is composed of a set of balanced items and a set of unbalanced items- [40,44,47]. For example, in the method proposed by Lorenzo-Seva & Ferrando for unbalanced multidimensional scales, acquiescence response variance is isolated in an independent factor, thus eliminating the variance due to acquiescent response from all the items, both from the balanced and the unbalanced subsets [44].

In the case of quasi-balanced or unbalanced scales, there is the method used in this study, the RIFA; a method that has been used by various authors for modeling the influence of acquiescence bias [40,47]. Comparing the RIFA with a method based on exploratory factor analysis, it has been found that the RIFA is more precise, robust, and less variable across conditions, especially in low balance scales [40]. In fact, the results of the present study show that this method makes it possible to control the acquiescence bias if the factorial score of the scale already decontaminated from this bias is used.

Finally, a general procedure that can be used to simultaneously control for acquiescence and social desirability bias is to identify a factor related to social desirability by using items that are considered indicators of this response bias. Then the loadings of the content items on this social desirability factor are used to compute a residual inter-item correlation matrix free of social desirability. Subsequently, the residual correlation matrix is analyzed by applying the Lorenzo-Seva & Ferrando method, which removes from the content factor those items whose variance is due to acquiescent responding [39,44]. This process makes it possible to analyze a residual inter-item correlation matrix that is free of the distortions caused by social desirability

and acquiescence and can be used in classical exploratory factor analysis to determine the factor structure of the questionnaire. Nevertheless, exploratory factor analysis can be run on each residual matrix and the factor structures can be compared without controlling for bias, controlling only for acquiescence, controlling only for social desirability, and controlling for both biases [38].

As future directions, it is considered relevant to carry out new both simulation and real data studies in which the performance of other newer procedures is evaluated.

Among the limitations of this study, it should be noted that the sample was made up mainly of workers from private organizations, the majority from the education sector and with a high level of formal education. This could limit the external validity of the study and implies that caution is necessary when generalizing the results obtained to populations of workers from public organizations, from other economic sectors, and with lower educational levels. On the other hand, although from the point of view of statistical requirements, the sample size was adequate, it is undoubtedly advisable to continue studies on the IWPQ with larger samples, in order to confirm the impact of acquiescence on its factorial structure.

Finally, given that social desirability is another bias that may also be relevant for instruments such as the IWPQ, in which a large part of the items reflect highly desirable behaviors, it is recommended to continue the investigation considering this bias as well, in order to assess to what extent the acquiescence and the social desirability simultaneously affect the factorial structure of the instrument.

## Supporting information

**S1 File.**
(SAV)

## Author Contributions

**Conceptualization:** Zuleima Santalla-Banderali, Jesús M. Alvarado.

**Data curation:** Zuleima Santalla-Banderali.

**Formal analysis:** Jesús M. Alvarado.

**Investigation:** Zuleima Santalla-Banderali.

**Methodology:** Zuleima Santalla-Banderali, Jesús M. Alvarado.

**Project administration:** Zuleima Santalla-Banderali.

**Resources:** Zuleima Santalla-Banderali.

**Supervision:** Zuleima Santalla-Banderali.

**Visualization:** Zuleima Santalla-Banderali, Jesús M. Alvarado.

**Writing – original draft:** Zuleima Santalla-Banderali.

**Writing – review & editing:** Zuleima Santalla-Banderali, Jesús M. Alvarado.

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
