## [Decision Letter · Decision Letter 0]

10 May 2022

PONE-D-22-05052Factorial structure of Individual Work Performance Questionnaire (Version 1.0) revisited: Evaluation of acquiescence biasPLOS ONE

Dear Dr. Santalla-Banderali,

Thank you for submitting your manuscript to PLOS ONE. After careful consideration, we feel that it has merit but does not fully meet PLOS ONE’s publication criteria as it currently stands. Therefore, we invite you to submit a revised version of the manuscript that addresses the points raised during the review process.

We look forward to receiving your revised manuscript.

Kind regards,

Ali B. Mahmoud, Ph.D.

Academic Editor

PLOS ONE

Journal Requirements:

Reviewers' comments:

Reviewer's Responses to Questions

**Comments to the Author**

1. Is the manuscript technically sound, and do the data support the conclusions?

Reviewer #1: Yes

Reviewer #2: Yes

2. Has the statistical analysis been performed appropriately and rigorously? 

Reviewer #1: I Don't Know

Reviewer #2: Yes

3. Have the authors made all data underlying the findings in their manuscript fully available?

Reviewer #1: Yes

Reviewer #2: Yes

4. Is the manuscript presented in an intelligible fashion and written in standard English?

Reviewer #1: Yes

Reviewer #2: Yes

5. Review Comments to the Author

Reviewer #1: The objective of this study was to evaluate the factorial structure of the Spanish version of the Individual Work Performance Questionnaire (Version 1.0), and to estimate the impact that acquiescence has on it as response bias. The introduction is well written and includes many references to literature supporting the hypothesis. The methods section is also described adequately (as far as I can judge, although I wasn’t familiar with the RIFA method). The research was conducted among 500 workers from different organizations in Ecuador, many working in the education sector. The authors conclude that a one-dimensional model is preferable.

I have several comments and recommendations that I feel would improve the article further;

- It is good that the impact of acquiescence bias is being investigated. I would ask the authors to describe acquiescence bias a bit further (line 209-213). Also please indicate that this is also called ‘agreement bias’. What does it mean practically? How often does this occur?

- There are many types of biases in self-report measures, such as social desirability bias and leniency effects (rating oneself favorably). Why do the authors only look at acquiescence bias? Would it not be better to correct for several types of biases, if possible? The authors could elaborate on this in the introduction. Why use self-report at all (there must be some advantages)?

- Results section, line 469: the explained variance of the models (respectively 12.1% and 6.7%) is rather low. Do you agree? Is this comparable to other research findings?

- It would be good if the authors could mention in the discussion some limitations of their research. For example, what about the sample characteristics and sample size? Wouldn’t it be more reliable to test the impact of acquiescence bias in a larger sample? And what about the external validity of the findings, in different sectors of work, or in different languages?

- Currently, the scores of the IWPQ are calculated for the three separate dimensions. In their article on the development of the IWPQ (Koopmans et al., 2013), the developers explicitly state that they do not recommend to calculate one total score because this is methodologically incorrect. Is your recommendation to reverse this original recommendation, and start working with one total score? And not three separate scores? Or both? Perhaps the authors could elaborate on this in the second to last paragraph (line 597).

- Is there a way to minimize acquiescence bias? It would be interesting to include this in the discussion.

Reviewer #2: This article clearly contributes to the advancement of knowledge about the problem of a individual work performance. There is degree of novelty in this endeavor. This is a very interesting topic to address. It’s a well-developed work that can be published. The sample size for safe conclusions is very adequate. The statistical analysis is certain. Results section is very good and clear. There is a good flow of discussion. The paper is well structured and written. The article adheres to the criteria of publication. It is original, and the results have never been published elsewhere. The article meets the required standards in terms of integrity and ethics.

6. PLOS authors have the option to publish the peer review history of their article (what does this mean?). If published, this will include your full peer review and any attached files.

Reviewer #1: **Yes: **Dr. Linda Koopmans

Reviewer #2: No

---

## [Author Response · Author response to Decision Letter 0]

31 May 2022

Manuscript PONE-D-22-05052

Dear Ali B. Mahmoud, Ph.D.

Academic Editor

PLOS ONE

Thank you for giving us the opportunity to submit a revised draft of the manuscript "Factorial structure of Individual Work Performance Questionnaire (Version 1.0) revisited: Evaluation of acquiescence bias" for publication in PlosOne. We appreciate the time and effort that you and the reviewers dedicated to providing feedback on our manuscript and are grateful for the insightful comments on and valuable improvements to our paper. We have incorporated the suggestions made by you and the reviewers. Changes are highlighted within the manuscript. Please see below for a point-by-point response to the comments. All page numbers refer to the revised manuscript file.

Comments to the authors of "Factorial structure of Individual Work Performance Questionnaire (Version 1.0) revisited: Evaluation of acquiescence bias" and authors' responses (in blue):

Journal Requirements

1. Please ensure that your manuscript meets PLOS ONE's style requirements, including those for file naming

We have carefully revised the manuscript, in order to correct the small mismatches that existed with respect to PLOS ONE's style. For this we have been guided by the following documents: 

2. Please review your reference list to ensure that it is complete and correct. 

We have reviewed the list of references, ensuring that it is complete and conforms to what is indicated in https://www.nlm.nih.gov/bsd/uniform_requirements.html#item36

Reviewer #1:

Many thanks to Dr. Linda Koopmans for taking the time to review our paper and for her invaluable comments. We hope to have answered them satisfactorily.

1. I would ask the authors to describe acquiescence bias a bit further (line 209-213). Also please indicate that this is also called ‘agreement bias’. What does it mean practically? How often does this occur?

In the introduction we have expanded on the acquiescence bias, indicating the following (new lines 215-230):

Response bias can affect the reliability estimates and the factor structure of questionnaires because it distorts the inter-item correlation matrix pattern, which in turn may have a considerable impact on the resulting factor structure [12, 38]. A review of the literature indicates that two of the main response biases that occur when measuring non-cognitive characteristics are social desirability (i.e., the tendency for people to present themselves in a generally favorable fashion) and acquiescence [12, 13, 38, 39] or agreement bias [11].

In the present study, we have focused on the acquiescence bias because, on the one hand, the distortion due to this bias seems to be bigger than the distortion due to social desirability and, therefore, its control may be more important when evaluating the psychometric properties of measurement instruments [38]. On the other hand, this bias frequently occurs when used self-reported scales [2] in which Likert-type scales are used [11-13] with direct items (i.e., the items are keyed in such a way that agreement results in a higher scale score) and reverse items (i.e., agreement with an item results in a lower score), as is the case of the IWPQ, and its possible incidence has not yet been considered in any of the studies aimed at evaluating the psychometric properties of IWPQ.

Likewise, we have included (new lines 234-241) an example that we consider helps the reader to understand what this bias means:

For example, in the case of the subscale of counterproductive behavior of the IWPQ in which all the items are inverted, acquiescence would be the tendency of people to select the response options identified with high numbers -indicative of a high frequency of occurrence of the behavior and, therefore, worse performance- due to the fact that the person does not pay attention to the substantive content of the item and gets carried away by the fact that in the vast majority of the items of the instrument the response options identified with high numbers are indicative of good performance.

2. There are many types of biases in self-report measures, such as social desirability bias and leniency effects (rating oneself favorably). Why do the authors only look at acquiescence bias? Would it not be better to correct for several types of biases, if possible? The authors could elaborate on this in the introduction. Why use self-report at all (there must be some advantages)?

Why do the authors only look at acquiescence bias? 

In this new version, in lines 222-230 we integrate into the same paragraph the reasons why we focused on the study of the impact of the acquiescence bias, which appeared scattered throughout the introduction.

In the present study, we have focused on the acquiescence bias because, on the one hand, the distortion due to this bias seems to be bigger than the distortion due to social desirability and, therefore, its control may be more important when evaluating the psychometric properties of measurement instruments [38]. On the other hand, this bias frequently occurs when used self-reported scales [2] in which Likert-type scales are used [11-13] with direct items (i.e., the items are keyed in such a way that agreement results in a higher scale score) and reverse items (i.e., agreement with an item results in a lower score), as is the case of the IWPQ, and its possible incidence has not yet been considered in any of the studies aimed at evaluating the psychometric properties of IWPQ.

Why use self-report at all (there must be some advantages? 

In the sub-section “Measurement of individual work performance: Individual Work Performance Questionnaire”, we have incorporated some of the advantages associated with self-report measures (new lines: 127-134):

Lastly, the instrument makes it possible to measure self-assessed work performance, which has the advantage of being able to have a performance evaluation in those cases in which it is impossible or extremely difficult to have objective measures of performance [31]. On the other hand, the evaluation that workers make of their own performance could be more accurate because they have more opportunities to observe their own performance than, for example, their supervisors or co-workers, especially considering that the evaluations carried out by supervisors and co-workers can be biased by the halo effect [31].

3. Results section, line 469: the explained variance of the models (respectively 12.1% and 6.7%) is rather low. Do you agree? Is this comparable to other research findings?

According to the criteria of Cohen [59] and Schäfer & Schwarz [60] we could consider that the size of the effect of the models we tested to assess the extent to which work performance and acquiescence scores are predicted by the sex, age, education level, time in the organization, time in current position, years of work experience, type of organization in which respondents work, and type of employee, was medium in the case of work performance and small in the case of acquiescence. This information was included in lines 495-496 and 503.

In the paper, we do not discuss whether the results obtained coincide with those found by other researchers, because the purpose of testing these models was only to exemplify how, if the acquiescence bias was not controlled, risks overestimating or underestimating the relationship between the IWPQ and other measures or criteria. This, in turn, can lead to inappropriate conclusions in the case of investigations, and to inappropriate organizational decision-making that could harm, not only the organizations but also the people who work in them, especially considering that organizations make administrative decisions on the basis of the evaluations they carry out of the performance of the workers.

4. It would be good if the authors could mention in the discussion some limitations of their research. For example, what about the sample characteristics and sample size? Wouldn’t it be more reliable to test the impact of acquiescence bias in a larger sample? And what about the external validity of the findings, in different sectors of work, or in different languages?

At the end of the Discussion section we have included the limitations of the study (new lines: 688-700):

Among the limitations of this study, it should be noted that the sample was made up mainly of workers from private organizations, the majority from the education sector and with a high level of formal education. This could limit the external validity of the study and implies that caution is necessary when generalizing the results obtained to populations of workers from public organizations, from other economic sectors, and with lower educational levels. On the other hand, although from the point of view of statistical requirements, the sample size was adequate, it is undoubtedly advisable to continue studies on the IWPQ with larger samples, in order to confirm the impact of acquiescence on its factorial structure.

Finally, given that social desirability is another bias that may also be relevant for instruments such as the IWPQ, in which a large part of the items reflect highly desirable behaviors, it is recommended to continue the investigation considering this bias as well, in order to assess to what extent the acquiescence and the social desirability simultaneously affect the factorial structure of the instrument.

5. Currently, the scores of the IWPQ are calculated for the three separate dimensions. In their article on the development of the IWPQ (Koopmans et al., 2013), the developers explicitly state that they do not recommend to calculate one total score because this is methodologically incorrect. Is your recommendation to reverse this original recommendation, and start working with one total score? And not three separate scores? Or both? Perhaps the authors could elaborate on this in the second to last paragraph (line 597).

In this new version in lines 611-622, we have extended the explanation regarding the recommendations derived from the results obtained in our study. In this sense, we have tried to explain more clearly that, according to our results, if acquiescence is not controlled, it seems to be justified to use the score obtained in each of the three IWPQ factors; but, controlling for acquiescence (which is what we recommend doing in all cases), our results suggest that the instrument is one-dimensional or essentially one-dimensional. This leads to recommending the use of the total score which is more useful and parsimonious in organizational contexts and applied studies.

In summary, in our study, we found that when acquiescence is not controlled, the three-correlated-factors model presented an acceptable fit; but, the model turns out to be conceptually conflicting when analyzing what has been observed regarding the correlations between the factors, and when considering the low reliability found on the specific factors.

At first, these results would validate the relevance of the three content domains of the construct, which leads to the appropriateness of using the scores of the factors. If the instrument is conceived as multidimensional, the use of the total score would not be admissible, unless essential unidimensionality is demonstrated; as occurred in the present study when controlling acquiescence. In this case, we found that the one-dimensional model and the essentially one-dimensional model showed goodness of fit. This makes it legitimate and advisable to use the total score obtained on the measurement [34], which is more useful and parsimonious in organizational contexts and applied studies.

6. Is there a way to minimize acquiescence bias? It would be interesting to include this in the discussion.

Following your suggestion, in this new version, we have incorporated some of the procedures available to minimize acquiescence bias (new lines: 637-687):

Thus, our results allow us to conclude that when the dimensionality of a job performance measure is evaluated, it is essential to use some correction method that allows response biases to be controlled, especially acquiescence. Otherwise, distorted factorial structures could be obtained that would lead to errors in the estimations of the scores, and the external validity of the measure [38, 63].

Regarding how to control the acquiescence bias, a procedure that can be used when the scale does not contain reverse items involves using the instrument to be evaluated and an additional questionnaire that contained direct and reverse worded items. An index is calculated that evaluates the degree to which people tend to respond acquiescently, and this index allows the variance due to acquiescence on an independent dimension to be isolated. The overall model should be tested using confirmatory factor analysis [64].

Now, the traditional way of trying to control acquiescence bias consists in developing the measuring instrument in such a way that it includes both direct and reversed items that have semantically opposite meanings so that they measure opposite poles of the same theoretical construct. Unfortunately, this strategy does not ensure that the means and covariance structure are free from acquiescence bias [47].

Due to the above, several “a posteriori” methods have been developed that allow for eliminating the impact of acquiescence using statistical procedures [63]. There are procedures based on an exploratory factor analytic approach [65] that can be used in cases of fully balanced scales -half of the scale items are worded in the opposite direction to the other half of the items-. 

There are also procedures for quasi-balanced scales -the scale has the same number of positive and negative items but with unequal saturations- or unbalanced scales -only a few items in the scale are worded in the opposite direction to the others, or the scale is composed of a set of balanced items and a set of unbalanced items- [40, 44, 47]. For example, in the method proposed by Lorenzo-Seva & Ferrando for unbalanced multidimensional scales, acquiescence response variance is isolated in an independent factor, thus eliminating the variance due to acquiescent response from all the items, both from the balanced and the unbalanced subsets [44].

In the case of quasi-balanced or unbalanced scales, there is the method used in this study, the RIFA; a method that has been used by various authors for modeling the influence of acquiescence bias [40, 47]. Comparing the RIFA with a method based on exploratory factor analysis, it has been found that the RIFA is more precise, robust, and less variable across conditions, especially in low balance scales [40]. In fact, the results of the present study show that this method makes it possible to control the acquiescence bias if the factorial score of the scale already decontaminated from this bias is used.

Finally, a general procedure that can be used to simultaneously control for acquiescence and social desirability bias is to identify a factor related to social desirability by using items that are considered indicators of this response bias. Then the loadings of the content items on this social desirability factor are used to compute a residual inter-item correlation matrix free of social desirability. Subsequently, the residual correlation matrix is analyzed by applying the Lorenzo-Seva & Ferrando method, which removes from the content factor those items whose variance is due to acquiescent responding [39, 44]. This process makes it possible to analyze a residual inter-item correlation matrix that is free of the distortions caused by social desirability and acquiescence and can be used in classical exploratory factor analysis to determine the factor structure of the questionnaire. Nevertheless, exploratory factor analysis can be run on each residual matrix and the factor structures can be compared without controlling for bias, controlling only for acquiescence, controlling only for social desirability, and controlling for both biases [38].

As future directions, it is considered relevant to carry out new both simulation and real data studies in which the performance of other newer procedures is evaluated.

---

## [Decision Letter · Decision Letter 1]

8 Jul 2022

Factorial structure of Individual Work Performance Questionnaire (Version 1.0) revisited: Evaluation of acquiescence bias

PONE-D-22-05052R1

Dear Dr. Santalla-Banderali,

We’re pleased to inform you that your manuscript has been judged scientifically suitable for publication and will be formally accepted for publication once it meets all outstanding technical requirements.

Kind regards,

Ali B. Mahmoud, Ph.D.

Academic Editor

PLOS ONE

Additional Editor Comments (optional):

Reviewers' comments:

Reviewer's Responses to Questions

**Comments to the Author**

1. If the authors have adequately addressed your comments raised in a previous round of review and you feel that this manuscript is now acceptable for publication, you may indicate that here to bypass the “Comments to the Author” section, enter your conflict of interest statement in the “Confidential to Editor” section, and submit your "Accept" recommendation.

Reviewer #1: All comments have been addressed

Reviewer #2: All comments have been addressed

2. Is the manuscript technically sound, and do the data support the conclusions?

Reviewer #1: Yes

Reviewer #2: Yes

3. Has the statistical analysis been performed appropriately and rigorously? 

Reviewer #1: Yes

Reviewer #2: Yes

4. Have the authors made all data underlying the findings in their manuscript fully available?

Reviewer #1: Yes

Reviewer #2: Yes

5. Is the manuscript presented in an intelligible fashion and written in standard English?

Reviewer #1: Yes

Reviewer #2: Yes

6. Review Comments to the Author

Reviewer #1: I want to thank the authors for their thorough work in addressing the comments made on their manuscript. They were handled and explained very well. I would recommend to publish this article.

Reviewer #2: The reviewed article significantly contributes to the advancement of knowledge about the issue of an individual work performance in the context of work efficiency. The level of novelty in this endeavor is noticeable. The findings regarding unproductive behavior and its relationship to performance in a Spanish context are compelling. The reviewed article is extensive, particularly taking into consideration the comments of the first reviewer. The sample size is adequate in order to formulate credible conclusions. The process of the statistical analysis has been conducted accurately. Results section is written correctly. There course of a discussion is well-organized. The article is well-structured. It adheres to the criteria of publication. Such results have never been published before, thus it constitutes a meritorical example of research paper. The article meets the required standards in terms of integrity and ethics.

7. PLOS authors have the option to publish the peer review history of their article (what does this mean?). If published, this will include your full peer review and any attached files.

Reviewer #1: **Yes: **Dr. Linda Koopmans

Reviewer #2: No

---

## [Editor Report · Acceptance letter]

12 Jul 2022

PONE-D-22-05052R1 

Factorial structure of Individual Work Performance Questionnaire (Version 1.0) revisited: Evaluation of acquiescence bias 

Dear Dr. Santalla-Banderali:

I'm pleased to inform you that your manuscript has been deemed suitable for publication in PLOS ONE. Congratulations! Your manuscript is now with our production department. 

Kind regards, 

on behalf of

Dr. Ali B. Mahmoud 

Academic Editor

PLOS ONE